# Presentation Attack Detection on Limited-Resource Devices Using Deep Neural Classifiers Trained on Consistent Spectrogram Fragments

**DOI:** 10.3390/s21227728

**Published:** 2021-11-20

**Authors:** Kacper Kubicki, Paweł Kapusta, Krzysztof Ślot

**Affiliations:** Institute of Applied Computer Science, Lodz University of Technology, Stefanowskiego 18/22, 90-001 Łódź, Poland; pawel.kapusta@p.lodz.pl (P.K.); krzysztof.slot@p.lodz.pl (K.Ś.)

**Keywords:** biometrics, presentation attack detection, mel-spectrogram, phoneme classification, deep neural networks

## Abstract

The presented paper is concerned with detection of presentation attacks against unsupervised remote biometric speaker verification, using a well-known challenge–response scheme. We propose a novel approach to convolutional phoneme classifier training, which ensures high phoneme recognition accuracy even for significantly simplified network architectures, thus enabling efficient utterance verification on resource-limited hardware, such as mobile phones or embedded devices. We consider Deep Convolutional Neural Networks operating on windows of speech Mel-Spectrograms as a means for phoneme recognition, and we show that one can boost the performance of highly simplified neural architectures by modifying the principle underlying training set construction. Instead of generating training examples by slicing spectrograms using a sliding window, as it is commonly done, we propose to maximize the consistency of phoneme-related spectrogram structures that are to be learned, by choosing only spectrogram chunks from the central regions of phoneme articulation intervals. This approach enables better utilization of the limited capacity of the considered simplified networks, as it significantly reduces a within-class data scatter. We show that neural architectures comprising as few as dozens of thousands parameters can successfully—with accuracy of up to 76%, solve the 39-phoneme recognition task (we use the English language TIMIT database for experimental verification of the method). We also show that ensembling of simple classifiers, using a basic bagging method, boosts the recognition accuracy by another 2–3%, offering Phoneme Error Rates at the level of 23%, which approaches the accuracy of the state-of-the-art deep neural architectures that are one to two orders of magnitude more complex than the proposed solution. This, in turn, enables executing reliable presentation attack detection, based on just few-syllable long challenges on highly resource-limited computing hardware.

## 1. Introduction

Remote biometric user verification becomes the predominant access control technology, due to the widespread use of mobile devices and attempts to develop convenient, yet reliable ways for securing access to resources and services [1]. A multitude of biometric traits have been successfully considered for identity resolution from data captured by mobile device cameras (face appearance, palm shape and papillary ridges, ear shape) and microphones (voice) [2]. Both sources of information can be used in a complementary, multi-modal recognition scheme, with the significance of individual sources weighted by input data quality. However, the unsupervised context of remote verification brings a severe threat of attacks against the data acquisition phase of the biometric data processing pipeline, executed by presenting spoofed or manipulated input. A natural means for presentation attack detection (PAD) in the case of voice modality, i.e., in a speaker authentication context, is a Challenge–Response (CR) scheme that attempts to validate the uttering of some system-prompted phrases [3]. Utterances to be generated should be short, to make liveness detection fast and unobtrusive, and hard to predict, to resist replay attacks, so random syllable sequences make good candidates for a challenge.

Effective PAD for voice biometrics relies on ensuring high phoneme recognition accuracy. Phoneme identification in audio waveforms, which is the first task of any Continuous Speech Recognition (CSR) procedure, has attracted an enormous amount of attention and multiple paradigms have been proposed over decades of intense research in the field [4]. However, only relatively recent breakthroughs brought by the advent of Deep Learning and Deep Neural Network concepts, advanced phoneme recognition performance to a level required for successful applications in a variety of real-world contexts [5]. One of the main approaches to phoneme-classification is to represent speech as images (Spectrograms or Mel-Spectrograms) and analyze these images using Deep Convolutional Networks [6]. The remarkable performance of state-of-the-art deep learning algorithms (with correct classification rates at the order of 85% for 39-class recognition in the case of spoken English language [7]), unfortunately, requires very complex architectures, which are ill-suited for implementation on resource-limited, mobile or embedded devices. Therefore, to enable protection of speaker authentication against presentation attacks in such a context, accurate and lightweight algorithms need to be sought.

The presented paper proposes a Challenge–Response procedure for counteracting presentation attacks on voice biometrics that is aimed at implementation on resource-limited computational architectures. To retain high attack detection accuracy, while enabling a significant reduction in the complexity of deep neural classifiers involved in processing, we propose to simplify the task of deep model training. We show that appropriate selection of training material, aimed at minimizing within-class data scatter, improves data analysis accuracy, thus providing a better use of the reduced capacity of small networks. As Convolutional Neural Networks, operating on windows of speech Mel-Spectrograms, are selected as a phoneme classification tool, the proposed idea is implemented by training the network only on spectrogram portions extracted from the central regions of phoneme articulation intervals.

The proposed training example selection scheme, which we refer to as a *central-window* scheme (as opposed to the commonly used sliding-window scheme), was evaluated using the TIMIT speech database [8], which is one of the most frequently used speech recognition benchmarks. To ensure a comprehensive assessment, we have considered a wide variety of combinations of speech preprocessing parameters—different frame lengths, spectrogram window sizes, and a number of Mel-frequency filters. We show that networks trained using the central-window scheme outperform, by a wide margin of 5 to 13 percent, networks trained using a sliding-window approach, for all considered combinations of speech preprocessing parameters.

Adopting the proposed approach enables achieving almost 74% spectrogram-window classification accuracy for the considered 39-class recognition problem even by using extremely compact networks, comprising 40–50 thousand parameters. This in turn makes the PAD procedure feasible for implementation in resource-constrained hardware: accurate spectrogram-window classification, which is the most complex PAD component, is supplemented with a computationally moderate speech preprocessing and a computationally inexpensive phoneme identification step. We show that the proposed PAD algorithm that uses compact CNN classifiers trained using the central-window scheme enables achieving low, 23% Phoneme Error Rates (PER).

As the considered neural architectures are very simple, we additionally examine the performance of ensembles of the derived classifiers. We find that spectrogram-window classification accuracy can be further improved, reaching almost 80%, when applying three-component ensembles. Although ensembling implies a multiple-fold increase in the network’s complexity, the resulting architecture comprising 150k parameters remains almost two orders of magnitude smaller than commonly used compact deep architectures, such as e.g., ResNet-18, while offering comparable classification accuracy. Having derived a resource-friendly, yet accurate phoneme recognition algorithm, we finally show that its application to the verification of prompted texts enables presentation attack detection based on few-syllable utterances, with over 99% confidence.

The structure of the paper is as follows: after a brief review of related concepts, focusing on state-of-the-art phoneme recognition methods and PAD algorithms for voice biometrics (Section 2), we explain in detail the proposed liveness detection procedure (Section 3) emphasizing the proposed central-window scheme for training set selection (Section 3.2) and the proposed approach to phoneme identification (Section 3.3). Results of experimental verification of the concept are presented and discussed in Section 4 and concluded in Section 5.

## 2. Related Work

The Challenge–Response (CR) utterance verification procedure, where a biometric system validates the uttering of some prompted texts, is conceptually the simplest presentation attack detection scheme against speaker recognition algorithms. Despite its shortcomings, it offers performance that is acceptable for a wide range of practical applications. Although it has been pointed out that the CR scheme is vulnerable to sophisticated attacks that use advanced real-time speech synthesis algorithms [9], alternative approaches can also be circumvented or require specialized equipment. For example, the effective replay attack countermeasure, reported by Sahidullah et al., requires the use of a special throat microphone sensor [10]. In addition, the *VAuth* authentication system developed by Feng et al. relies on a wearable device that detects body-surface vibrations which accompany speech [11]. Another approach to detecting attacks against voice biometrics—a method proposed by Zhang’s et al. [12]—assumes the presence of two microphones, as it is based on time-arrival difference measurements. The alternative introduced by Wang in et al. [13] relies on detecting characteristic breathing patterns coexisting with speech; however, it requires holding a microphone in close proximity to the mouth. Taking into account the drawbacks of the presented PAD ideas, it is clear that the application of a basic CR scheme in remote voice biometrics seems to be well justified.

To make CR-based validation of user authenticity unobtrusive and reliable, prompted texts need to be short, which implies a need for highly accurate speech analysis. Meeting this objective has become possible due to recent breakthroughs in deep learning. Since Dahl et al.’s seminal paper presenting a hybrid deep neural network—hidden Markov model (DNN-HMM) [14]—many diverse directions of utilizing deep learning for speech recognition have been explored. One of the possibilities is usage of Recurrent Neural Networks (RNNs)—a tool designed specifically for the purpose of sequence analysis. Graves et al. reported a 17.7% phone error rate (PER) on the TIMIT database [15] in 2013. Even better results—14.9% PER (by far the best for TIMIT) was obtained by using the RNN architecture proposed in [16] and composed of Light Gated Recurrent Units (Li-GRU) (totaling 7.4 million parameters). A different approach utilizes well-established Convolutional Neural Networks (CNNs), where a speech signal is transformed into a spectrogram. Abdel-Hamid et al. first proposed this concept and reported 20.17% PER, again on the TIMIT benchmark [6]. Later, a hierarchical CNN using max-out activation function has been proposed, and achieved a 16.5% phoneme error rate [17]. In 2020, Gao et al. used U-Net architecture with 7.8 million parameters adopted from the semantic image segmentation task and reported 19.6% PER [18].

Unfortunately, excellent recognition rates offered by deep neural network classifiers require complex architectures that involve several millions parameters, which is problematic for efficient implementation on resource-limited devices. Less complex classification methods, such as Support Vector Machines (SVM) and Random Forests, have also been proposed—e.g., by Ahmed et al., who reported an SVM RBF classifier, 8 times faster and 153 times lighter (with respect to feature size) than the state-of-the-art CNN solution [19]—nevertheless, they are still outperformed by deep learning based methods.

By far, algorithms for phoneme recognition in continuous speech have not been tailored to the realm of limited-resource devices. Instead, to balance usability and hardware limitations, isolated word recognition a.k.a. Keyword Spotting has been considered, and several efficient deep learning algorithms were proposed to do the task. For example, works presented by Sainath and Parada [20] in 2015, Tang and Lin [21] in 2017, and Anderson et al. [22] in 2020 demonstrate compact CNN architectures trained on Mel-spectrograms of short audio files for keyword recognition. The number of network parameters presented in those papers varies from 1.09M in the case of Sainath and Parada’s *tpool2* network, through 131K of Anderson’s et al. *kws2*, down to even 19.9K in the case of *res8-narrow* architecture proposed by Tang and Lin, whilst achieving over 90% test accuracy on the Google Speech Commands benchmark dataset [23], proving that small-footprint CNNs can be successfully utilized in audio recognition tasks.

## 3. Liveness Detection Procedure

A scheme for Challenge–Response based presentation attack detection, which is considered in the presented research, is to generate random texts that are to be uttered by a speaker and subsequently validated by the algorithm. The proposed procedure, depicted schematically in Figure 1, comprises three main data processing phases. The first one—data preprocessing (a block denoted by ‘P’)—converts the input speech signal into a series of overlapping Mel-Spectrogram windows, which in the second phase are analyzed using an appropriately trained Convolutional Neural Network (a ‘CNN’ block). A sequence of labels, predicted for subsequent windows, is then examined to identify the uttered phonemes (a ‘PI’ block). The detected sequence is finally confronted with the expected outcome and a decision on the procedure outcome (i.e., whether the test was passed or it failed) is made.

### 3.1. Speech Signal Preprocessing

Utterances are transformed into Mel-Spectrograms using a standard speech signal preprocessing procedure. First, the input audio samples are split into a sequence of overlapping frames. Next, each frame, after being adjusted with a window function, is subject to the Discrete Fourier Transformation. Finally, the resulting magnitude spectra are wrapped around a bank of triangular filters centered at a set of Mel-scale frequencies. To ensure a comprehensive evaluation of the proposed concept, different combinations of key parameters of the adopted signal preprocessing procedure were considered. We examined different frame lengths, which for a given sampling frequency, determine the range of spectral signal composition. Furthermore, we considered different numbers of Mel-filters, which determine the spectral resolution of the analysis. Lastly, we examined different Mel-Spectrogram window lengths, which determine the amount of contextual information considered in label prediction.

### 3.2. Derivation of Spectrogram-Window Classifier

Examples that are used for the training of CNN phoneme classifiers are typically collected by sliding some fixed-width analysis window over subsequent spectrogram regions, with some fixed overlap (shown schematically in Figure 2). This strategy provides the network with comprehensive information on spectrogram structures that represent the articulation of feasible phone combinations. However, if the network’s size, and thus, its information storage capacity (that can be coarsely estimated, e.g., using Vapnik–Chervonenkis’ dimension [24]) decreases, the high within-class variability of structures can no longer be correctly captured. Therefore, to simplify the task to be learned, we postulate to limit the training examples only to a subset of patterns that are maximally consistent. This way, all considered models can specialize in learning the most salient, class-specific spectrogram structures. To meet this objective, class examples are represented only by spectrogram regions that are located within the central intervals of phone-articulation periods, where the patterns are similar to each other and where spectrogram window contents are less affected by highly variable contextual information (see the bottom part of Figure 2). We define the aforementioned ’central interval’ as a region that covers up to five percent of a given phoneme articulation period, centered around the period’s midpoint.

Given the training sets generated using two different approaches (either a conventional sliding-window based method or examples selected using the central-window scheme), one can proceed with searching for a compact CNN architecture to perform the window-labelling task. This target architecture is constrained only by two factors. The first one is the shape of the input data: a two-dimensional matrix of size w×h, where *w* is a width of the considered Mel-Spectrogram window and *h* denotes the number of Mel-filters (i.e., the number of spectral components that represent each frame). The second constraint is the number of classes considered in the recognition, and it determines the number of the network’s outputs.

As the considered neural classifier is a nonlinear algorithm with a considerable number of parameters, non-gradient optimization methods seem to be feasible candidates for its architecture optimization. Of the many possible candidate algorithms, a Nelder–Mead’s simplex method [25] was adopted as a tool for task realization. However, before applying the optimization, two additional constraints on the target architecture were imposed. The first one applies to the feature extractor structure, whereas the second one to the structure of the dense part of a CNN. In the former case, we fixed the number of convolutional layers to either four or five, which bounds the maximum scaling of relatively small input images (we assume that convolutions are followed by downsampling). In the latter case, we fixed the number of dense layers to three, to enable the formation of arbitrary decision region shapes. The optimization objective was to maximize the classification accuracy and the expected optimization outcome was a selection of compact CNN architectures that satisfy this objective.

### 3.3. Identification of Uttered Phonemes

The CNN-based spectrogram window classifier produces a sequence of label predictions with a temporal resolution determined by a between-window shift. It follows that the labels assigned to subsequent windows that slide through a given phone should be the same, if the shifts are small enough, with the number of symbol repeats depending on the phoneme articulation duration. Therefore, the Phoneme Identification module (PI block, Figure 1) adopts the following rule for assembling window-label predictions to phonemes. The phoneme is recognized only if a sequence comprising a sufficiently long series of identical window-labels is found. The length of this sequence is determined by a threshold parameter that is experimentally selected based on phoneme duration statistics and phoneme recognition performance.

### 3.4. Estimation of Attack Detection Confidence

Utterance validation requires confronting the expected and actual phoneme sequences. As uttered text validation can be regarded as a series of independent experiments of testing pairs of corresponding phonemes for equality, presentation-attack detection confidence can be quantified based on the properties of binomial distribution. The probability of correct recognition of at least *k*-phonemes in an *n*-element sequence by a trained classifier is given by:(1)$$p_{c}^{\left(k,n\right)}=\sum_{i=k}^{n}C_{n}^{i}{\bar{p}}_{c}^{i}{\left(1-{\bar{p}}_{c}\right)}^{n-i}$$
where Cni=ni is a Binomial Coefficient and ${\bar{p}}_{c}$ denotes average probability of correct phoneme recognition. Assuming that the average phoneme-recognition probability ${\bar{p}}_{c}$ is higher than the average probability of a random match ${\bar{p}}_{rand}$, we define PAD confidence as a gain in probability of correct recognition of at least *k*-phonemes by means of the trained classifier ($p_{c}^{\left(k,n\right)}$) over a probability of getting this result by random guessing ($p_{rand}^{\left(k,n\right)}$):(2)$$C^{\left(k,n\right)}=p_{c}^{\left(k,n\right)}-p_{rand}^{\left(k,n\right)}$$

Given the expression (Equation 2), one can determine the parameters (length of a challenge sequence and the minimum required number of correctly recognized phonemes) that ensure obtaining some assumed minimum confidence level θ:(3)$$k^{*},n^{*}=arg\min_{n,k\leq n}\left(C^{\left(k,n\right)}-\theta \right)\geq 0$$

Observe that maximizing phoneme recognition accuracy ${\bar{p}}_{c}$ provides two major benefits. It enables shortening a challenge, making the liveness detection procedure more friendly, and, for a given challenge length, increases decision-making confidence.

## 4. Experimental Evaluation

The following main objectives were pursued throughout the experimental part of the presented research. The first and most important goal was to verify the hypothesis regarding a possible improvement in classification accuracy due to the adoption of the proposed central-window scheme for training example selection. The second goal of the experiments was to determine whether one can derive, based on a training set constructed using the proposed scheme, a compact CNN architecture, well-suited for implementation on hardware-limited devices, offering sufficiently high phoneme recognition accuracy. Finally, for the derived classification algorithms, we were interested in estimating the parameters of the challenge: Phoneme sequence length and the minimum number of matches, which are required to ensure the desired presentation attack detection confidence levels.

Throughout the experiments, we assumed 1 millisecond shifts between subsequent spectrogram windows, and we examined different combinations of speech preprocessing parameters: frame and window lengths and a number of filter banks. Specifically, we considered five different frame lengths: $n_{DFT}=160,256,512,1024,1600$, three Mel-Spectrogram window lengths: WL=64, 128, and 256 (as we chose 1 ms window shift, tested windows were, respectively, 64 ms, 128 ms and 256 ms-long), and two lengths of Mel-filter banks: $N_{FB}=40$ and 128.

### 4.1. TIMIT Speech Corpus

For all experiments, we used the TIMIT speech database—one of the best established speech data resources for research on automatic speech recognition. TIMIT contains recordings of 630 speakers, representing eight main American English dialects, with each speaker uttering 10 phonetically rich sentences for a total of 6300 sentences, or over 5 h of audio recordings. Although the original TIMIT corpus transcriptions are based on the set of 61 phonemes, following [26], it is widely agreed that some of the phonemes should be considered as the same class, resulting in mapping of the original phoneme labels set into 39 classes—a labeling scheme that we also follow. We used a randomly selected 10% of the TIMIT core training set as a validation dataset. Presented accuracies (ACC) and phoneme error rates (PER) have been calculated using the TIMIT core test set.

### 4.2. CNN Architecture Derivation

The first part of our experiments was concerned with the derivation of a compact architecture, suitable for mobile devices. As mentioned in Section 3.2, we assumed that CNN with either four or five convolutional and three dense layers would be a starting point for optimization procedures. Except for the output softmax layer, LeakyReLU [27] activation functions were used for all network’s neurons. Since at this point the proper speech preprocessing parameters were unclear, we decided to perform the optimization using input data of size 40 rows (i.e., 40 frequency components) and 256 columns (i.e., 256 millisecond-long spectrogram windows). The vector of hyperparameters that was used in the optimization comprised: the number of filters per each convolutional layer, together with the parameters of the filters’ structure (widths and heights), and an amount of neurons per each dense layer. The optimization objective was to maximize the classification accuracy on the validation dataset. In each iteration, CNN was trained using a categorical cross-entropy loss with L2-regularization. Table 1 presents architecture details and performance for both initial models and for the best models obtained after 100 optimization steps, as well as for the reference ResNet-18 network—the most compact variant of He’s et al. ResNet architectures [28].

As it can be seen from Table 1, both optimized architectures are significantly more complex than the initial ones, although the corresponding performance gains are rather minor. An interesting feature of these architectures is an increase in kernel size (beyond what is typical in visual object recognition [28,29,30]), which suggests the importance of broad contextual information in spectrogram-windows classification. As the classification accuracy differences between initial and optimized networks were not very large, we performed further experiments for all architectures. However, to facilitate the task, only initial simple five-convolutional layer architectures (of complexity ranging from 34,535 weights for input windows of shape 40 × 64 to 51,975 for networks analyzing 128-by-256 windows) were considered to determine the optimal signal preprocessing parameters. In each experiment, we used a well-established ResNet-18 architecture as a reference.

### 4.3. Evaluation of the Proposed Classifier-Training Scenario

To verify the hypothesis that training networks with data extracted only from central regions of phoneme articulation intervals improves phoneme classification accuracy, we run a set of experiments for different combinations of speech preprocessing parameters. We trained classifiers either on data prepared using the proposed central-window scheme or using a sliding-window approach [31]. For all tests, parameter optimization was made using Adam optimizer [32], with a fixed value of learning rate of 0.0003, batch size of 64, and weight initialization using a scheme proposed by Glorot et al. [33]. No pre-training or transfer learning techniques were used.

The results, summarized in Table 2, clearly show that the proposed training scheme outperforms the sliding-window based approach. For every combination of speech preprocessing parameters and for all tested architectures, the classification results for networks trained on central-windows are higher by 5 to 13 percent. Moreover, simple CNNs trained on central-region examples in most cases perform better than ResNet-18 trained using a sliding-window approach.

Analysis of different speech preprocessing parameter setups shows that an increase both in the spectral resolution of the speech representation and an increase in the amount of considered contextual information that is provided by longer analysis windows improves results. In the case of frame lengths, 256-long or 512-long sample sequences (16 ms or 32 ms for the 16 kHz TIMIT speech sampling frequency) seem to be optimal. Results presented in Table 2 were summarized in a graphical form in Figure 3 and Figure 4.

### 4.4. Phoneme-Sequence Identification

Classification accuracy evaluated for spectrogram windows is not a relevant metric for evaluating utterance recognition performance, as the expected speech recognition outcome needs to be expressed using correct phoneme identification rates. This requires assembling subsequent window-label predictions into predictions of uttered phoneme sequences and confronting these results with the ground truth. As it has been indicated in Section 3.3, a phoneme is identified if a sufficiently long sequence of its consecutive predictions is detected. The number of label repetitions needs to be large enough to avoid false predictions that could occur, e.g., during phone-articulation transients, but at the same time, it needs to ensure correct responses to short-duration phones (phone-duration distribution for TIMIT dataset is shown in Figure 5).

To determine the optimal amount of repeating window-labels that provides a reliable phoneme detection, we execute the following procedure. First, we choose the initial five-layer CNN architecture and we fix data-preprocessing parameters at values that proved the best in spectrogram-windows classification ($n_{DFT}=256$, WL=256, $N_{FB}=128$ and 1 ms-long hops between subsequent windows). Then, using the TIMIT training set, we begin an iterative search procedure. An initial value for the target label repetition threshold—$\theta_{0}$—addresses the aforementioned compromise between phone articulation transient effects and short-phone detection capability. We assume that $\theta_{0}=8$, i.e., we initially test the consistency of classification results for at least eight consecutive windows (with origins evenly distributed within the 8-millisecond interval). For each *i*-th iteration of the procedure, only sequences of at least $\theta_{i}$ identical, CNN-produced window-labels are considered as a successful phoneme recognition result and are assigned a phoneme-label. Given a ground truth—a sequence of phoneme labels manually assigned to the considered utterances—a Phoneme Error Rate score is then calculated, and the procedure is repeated for another candidate $\theta^{i+1}=\theta_{i}+1$.

The Phoneme Error Rate—PER—is defined as:(4)$$\begin{array}{c} PER=\frac{S+D+I}{N} \end{array}$$
where S,D,I denote the number of substitutions, deletions, and insertions that need to be made to map the produced phoneme-sequence onto the expected one, and *N* is the length of the reference sequence of phonemes in a challenge utterance.

Results of the threshold selection procedure have been presented in Figure 6. It can be seen that, for the considered dataset, an optimum range of threshold values can be clearly identified. Given the between-window shift of 1 ms, these values provide a balance between the preservation of short phoneme detection feasibility (such as, e.g., ‘b’, with a mean duration of 17 ms or ‘d’ with 21 ms mean length) and erroneous detections made mainly in transient regions.

Having set the phoneme detection threshold to n=15, we calculated PER score for the recognition of TIMIT core test set utterances for different considered architectures and all considered combinations of speech preprocessing parameters. In all cases, we used the same 39-class CNN classifiers as for the window contents recognition experiments presented in Table 2. The results, summarized in Table 3, further confirm superiority of the proposed training set selection scheme for simple CNN classifiers. In the case of ResNet-18, one can observe that the gains in accuracy are lower or even that the performance deteriorates. The main reason for this effect is a significant, approximately ten-fold reduction in volume of the training set generated using the central-window scheme (from over a million examples collected for the sliding-window scheme, to around 150 thousand examples). This reduction clearly impairs the learning of almost a dozen million-parameter ResNet-18 architecture. The best achieved value of PER for ResNet-18, trained on examples prepared using the central-window scheme is 18.67% (which is close to the performance of the state-of-the-art architectures). PER scores for simple CNNs varied from 33.4% for the most compact architecture, comprising around 34k parameters, to 24.4% for the architecture comprising 52k parameters. These results also confirm the significance of the amount of contextual information (the results improve as the window length increases), but provide no clear conclusions regarding frame length.

Detailed information on spectrogram-windows classification results, provided by the confusion matrix (Figure 7), is consistent with the work reported elsewhere [34]. The phoneme posing the greatest difficulty for a classifier is the vowel ‘uh’, which is notoriously confused with the vowels ‘ah’ and ‘ih’, whereas the highest recognition accuracy is obtained for short consonants (‘b’, ‘dx’, ‘q’).

To emphasize the differences caused by applying the two considered CNN training schemes, the results of a sample speech Mel-spectrogram fragment classification have been presented in Figure 8 and Figure 9. The plots drawn above the presented Mel-spectrogram, comprising seven phonemes (with phoneme articulation boundaries delimited with red vertical lines), depict the temporal evolution of the probabilities generated by the corresponding seven CNN outputs. It can be seen that the plots are qualitatively different, depending on the adopted training scenario. In case of the proposed central-window training scheme, there are clearly visible probability peaks that emerge in the central regions of phoneme articulation periods. On the other hand, the responses of a classifier trained using the sliding-window scheme are spread over the whole phoneme articulation period, but with lower and highly variable magnitudes.

Performance of the optimized five-convolutional layer architecture (see Table 1) trained using the central-window scheme (summarized in Table 4) shows that it is competitive with approximately seventeen times more complex ResNet-18 both in terms of window classification accuracy (81.25% compared with 81.94% for ResNet-18) and phoneme recognition (lowest obtained PER—22.91% compared with 18.67% for ResNet-18). Throughout the experiments, we used 256 ms-long spectrogram windows, both proposed numbers of Mel-filters (40 and 128) and all considered frame lengths ($n_{DFT}=160,256,512,1024,1600$).

### 4.5. Ensembling Simple Classifiers

Motivated by recent advances in ensembling deep classifiers [35], we also tested whether combining the simplest architectures that operate on different frame lengths: 128, 256, and 512 samples, thus analyzing different information, could improve phoneme recognition. We adopted a simple bagging approach with majority voting as a decision fusion strategy. Experiment results, summarized in Table 5, show that the lowest PER for the ensemble of ResNet-18 architectures equals 17.32% (as compared to 18.67% without ensembling) and 22.12% for an ensemble of the 5-layer-init architecture (compared to 24.40% without ensembling and 22.91% of its Nelder–Mead optimized variant).

Phoneme-recognition accuracy for individual classifiers and for classifier ensembles has been summarized in Figure 10, where the results are grouped according to the input data shape. It can be seen that, in any case, the application of classifier ensembles reduces phoneme error rates, compared to the mean performance of individual classifiers. Furthermore, one can see that the proposed central-window training scheme is superior for both individual simple architectures as well as for their ensembles. Finally, the performance of ensembles of simple networks, with complexity of the order of 150k parameters, gets close to the performance of ResNet-18 architecture, which is approximately fifty-times more complex.

### 4.6. Challenge Parameters Estimation

Given phoneme recognition accuracies, one can estimate the necessary challenge sequence length that ensures some assumed PAD confidence levels. For the considered attack scenario, where some prerecorded utterance of a legitimate user is provided as a response to the challenge, only random phoneme matches can occur. Although a phoneme random match probability for an *m*-phoneme alphabet is $\frac{1}{m}$, feasible utterances must be syllable-based, so, as the worst-case, we assume that ${\bar{p}}_{rand}=\frac{1}{m_{v}}$, where $m_{v}$ is the number of vowels. Taking this into account and assuming that challenges are generated as random syllable sequences, we provide challenge sequence length estimates for different assumed PAD confidence levels for different considered architectures, trained using a central-window scheme (Table 6).

### 4.7. Computational Complexity and Performance Results

Total computational complexity of the proposed solution is the sum of the preprocessing step complexity (derivation of Mel-spectrograms) and the complexity of CNN-classifier forward-pass execution. Assuming the adopted notation (where $n_{DFT}$ denotes frame length, $N_{FB}$ and WL denote number of Mel-filters and window-length, respectively), the complexity of the PAD procedure is low and can be estimated as: $O\left(n_{DFT}*\log\left(n_{DFT}\right)\right)+O\left(n_{DFT}\right)+O\left(N_{FB}*WL*\log\left(N_{FB}*WL\right)\right)$.

The proposed PAD procedure has been implemented on a mobile device. We used Xiaomi Pocophone F1 powered by a Qualcomm Snapdragon 845 processor, 6 GB of RAM, and Android 10.0 operating system. The two network architectures: the initial 5-layer classifier and ResNet-18, were converted to TensorFlow-Lite models with all variables represented as 32-bit floats. Classification of spectrogram-window of size 40 × 256 using the simple CNN architecture takes 6 ms, whereas the classification performed by Resnet-18 takes around 66 ms. Moreover, an order of magnitude lower are the memory requirements for model storage: around 1 MB is consumed by the simple CNN classifier compared to over 40 MB required by ResNet-18.

### 4.8. Discussion

The presented experiments confirmed the hypothesis that one can increase the accuracy of CNN-based phoneme classifiers by adopting the proposed central-window based training scheme. The superiority of this approach has been confirmed for all tested architectures: from the complex ResNet-18 network comprising several million parameters, through optimized CNN networks with several hundred thousand weights, to extremely simple, structures, comprising only several dozen thousand parameters (as shown in Table 2 and Table 4, presenting spectrogram-window classification results, and in Table 3 and Table 4, presenting phoneme recognition performance). One can also observe that performance gains grow with a reduction in the network’s complexity, which supports our conjecture concerning a better use of a limited network’s information capacity due to training on data with the reduced within-class scatter.

The ability to simplify CNN classifier structure without compromising recognition accuracy enables executing presentation-attack detection on resource-limited devices. We show that architectures with as few as 50k parameters trained using the central-window scheme provide higher spectrogram-window classification accuracy than several-million parameter ResNet-18 trained using the sliding-window scheme. We also show that ensembling these simple architectures provides further window-recognition and phoneme-recognition improvements while keeping the complexity of the resulting network at the order of 150k parameters.

Finally, the results provided in Table 6 show that Challenge–Response presentation attack detection can be successfully executed using the proposed simple architectures, providing high decision-making confidence and requiring almost identical challenge complexity as in the case of PAD executed using large networks.

## 5. Conclusions

The main contribution of the research reported in the paper is the experimental confirmation of the hypothesis that it is possible to improve CNN-based phoneme recognition accuracy by training a classifier on speech spectrogram windows that are extracted only from central regions of phoneme articulation intervals. This finding, which clearly needs to be verified on a variety of distinct speech corpora, has potential consequences for general speech recognition research, not only for presentation-attack detection that was the focus of the paper. The observed recognition gains grow as neural classifier complexity decreases, suggesting a better use of the architecture’s information capacity.

By using the proposed training scheme, we have also proven that it is feasible to develop reliable Challenge–Response based presentation-attack detection algorithms, which employ neural architectures of complexity that can be orders of magnitude lower than the commonly used ones. As a consequence, they can be easily implemented on mobile or embedded devices, providing high verification confidence even for short, few syllable-long prompted utterances, making the liveness detection procedure unobtrusive. Therefore, the proposed concept can become an attractive alternative to other existing PAD methods for voice-based user authentication that is to be executed under an unsupervised data acquisition scenario.

## Figures and Tables

**Figure 1 sensors-21-07728-f001:**
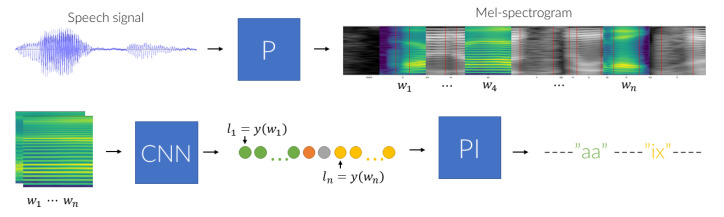
A diagram of the proposed utterance analysis method. The data preprocessing module (P) transforms input speech to Mel-Spectrogram windows (denoted by $w_{i}$), which are subsequently classified by a Convolutional Neural Network that assigns a label $l_{i}$ to each input window $w_{i}$. The resulting sequence of predicted labels is analyzed by a phoneme identification module (PI), which produces a sequence of recognized phonemes.

**Figure 2 sensors-21-07728-f002:**
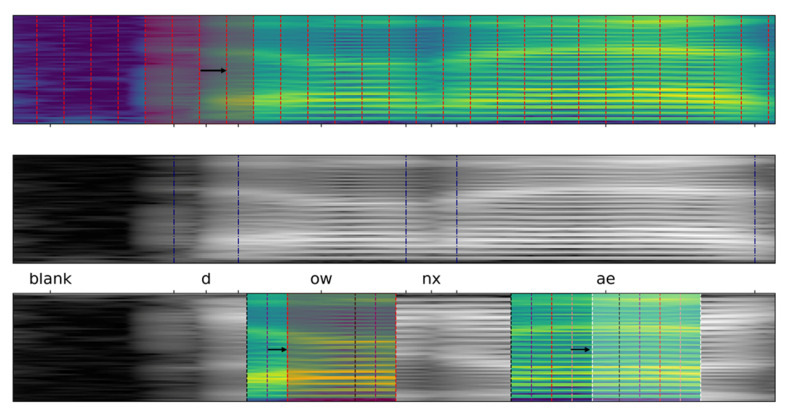
Mel-Spectrogram slicing scheme for building example sets for a conventional sliding window-based procedure (**top**) and for the proposed concept (**bottom**). In the latter case, windows are extracted only within a central region of an articulation period. Original spectrogram, together with phoneme annotations, is shown for reference in the (**middle**) row.

**Figure 3 sensors-21-07728-f003:**
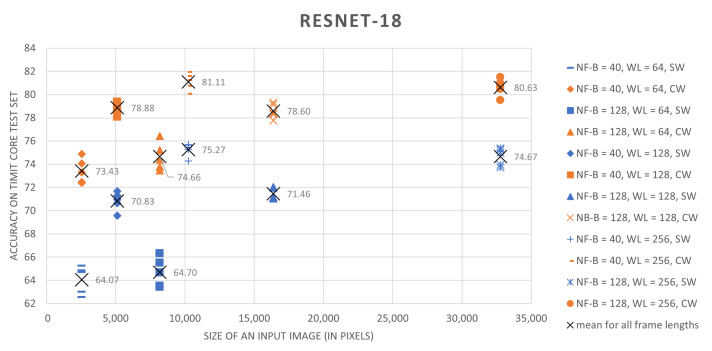
ResNet18 classification accuracies on the TIMIT core test set as a function of input image size. Models trained using datasets derived using a central-window scheme (CW) are shown in orange, while models trained on dataset prepared using sliding-window method (SW) are shown in blue. The presented point clusters correspond to different frame lengths.

**Figure 4 sensors-21-07728-f004:**
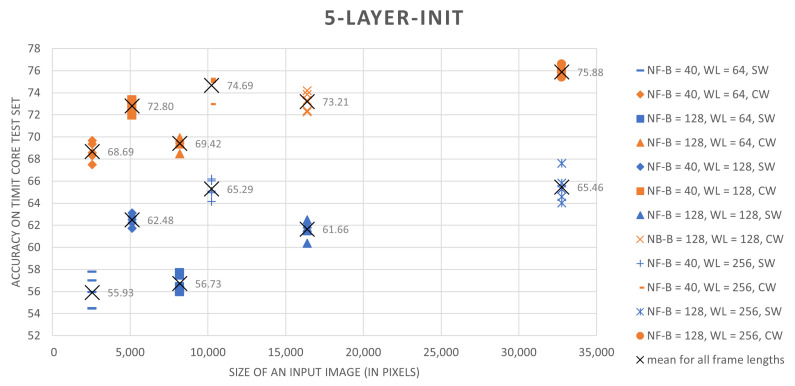
Classification accuracies for the initial 5-layer CNN architecture as a function of input image size and speech preprocessing parameters.

**Figure 5 sensors-21-07728-f005:**
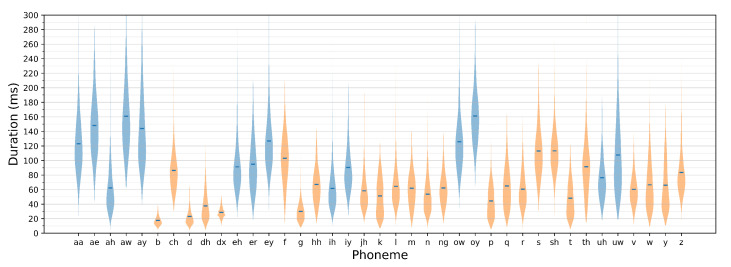
Violin plot showing duration of TIMIT core training set phonemes. Vowels are indicated in blue, while non-vowels in orange. Dark-blue ticks indicate mean phoneme durations.

**Figure 6 sensors-21-07728-f006:**
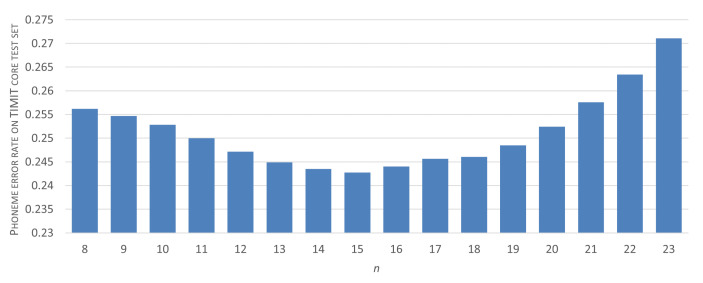
Distribution of Phoneme Error Rates for different phoneme-detection thresholds and a simple CNN classifier (central-window scheme, $N_{FB}=128$, WL=256, $n_{DFT}=256$).

**Figure 7 sensors-21-07728-f007:**
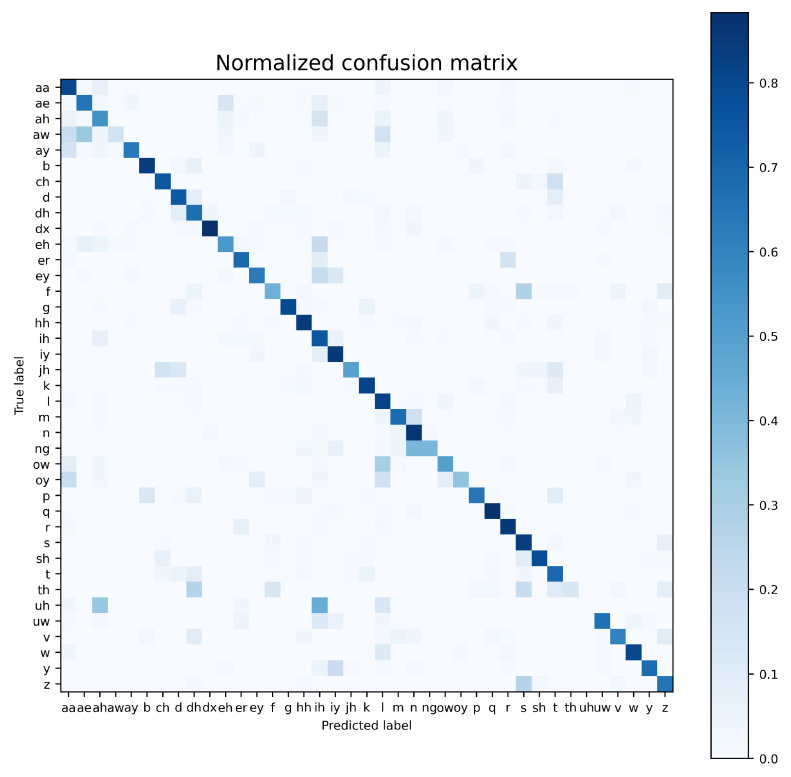
Confusion matrix for the best-performing simple CNN architecture ($N_{FB}=128$, WL=256, $n_{DFT}=256$), trained using the proposed, ‘central’ examples.

**Figure 8 sensors-21-07728-f008:**
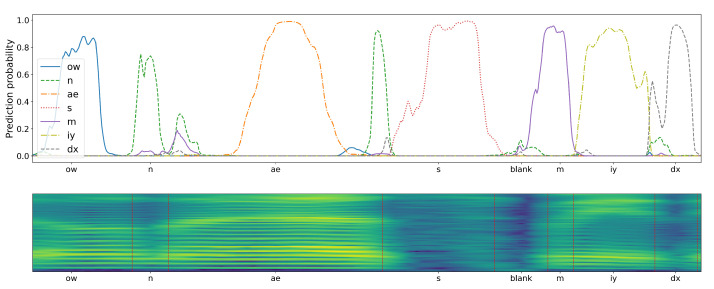
Probabilities produced at seven CNN-classifier outputs (**top**) during analysis of Mel-Spectrogram composed of seven different phonemes (**bottom**), for a classifier trained on central phoneme examples. Observe an erroneous detection of a consonant ‘n’ in the middle part of the plot.

**Figure 9 sensors-21-07728-f009:**
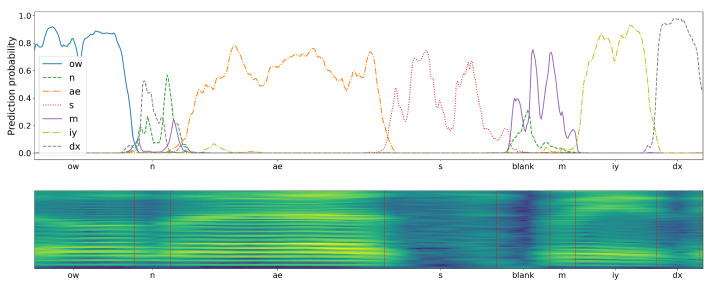
Probabilities produced for the same spectrogram as in Figure 8 by a classifier trained using the sliding-window scheme. Observe an erroneous detection of a consonant ‘dx’ in the leading part of the plot.

**Figure 10 sensors-21-07728-f010:**
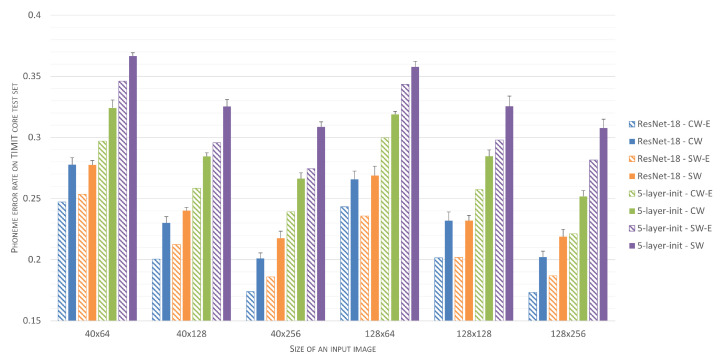
Phoneme error rates calculated for individual classifiers and their ensembles, determined for six groups of experiments, involving different shapes of input Mel-Spectrogram windows (either 40 or 128 rows and 64, 128, and 256 columns). Performance of Res-Net-18 and the derived CNNs, trained either with the proposed central-window scheme (suffix CW) or sliding window (suffix SW), is shown. For each experiment, three different frame-lengths (128, 256 and 512) were used to preprocess input speech. Mean performance for three individual classifiers is shown using solid bars with whiskers denoting 99% confidence intervals, whereas performance for three-element classifier ensembles is presented using dashed bars.

**Table 1 sensors-21-07728-t001:** Architectures, complexity (in number of parameters), and phoneme classification accuracy (ACC) for the initial and optimized CNN models, and for the ResNet-18 network.

Architecture					#Params	ACC (%)
	Convolutional Layers			Dense Layers		
	#layers	kernel size	#filters	#neurons		
4-layer-init	4	3, 3	32, 32, 32, 32	32, 32, 39	63,207	75.64
4-layer-opt	4	7, 7	11, 81, 46, 47	139, 94, 39	558,955	79.68
5-layer-init	5	3, 3	32, 32, 32, 32, 32	32, 32, 39	47,879	75.25
5-layer-opt	5	9, 9	9, 26, 28, 59, 74	106, 59, 39	637,816	79.93
ResNet-18					≈11 M	81.57

**Table 2 sensors-21-07728-t002:** Spectrogram-window classification accuracy (ACC) on the TIMIT dataset for different tested architectures and different combinations of speech preprocessing parameters.

Network Architecture	Central Window Method				Sliding Window Method				Network Architecture	Central Window Method				Sliding Window Method			
	$N_{\mathrm{FB}}$	WL	$n_{\mathrm{DFT}}$	ACC (%)	$N_{\mathrm{FB}}$	WL	$n_{\mathrm{DFT}}$	ACC (%)		$N_{\mathrm{FB}}$	WL	$n_{\mathrm{DFT}}$	ACC (%)	$N_{\mathrm{FB}}$	WL	$n_{\mathrm{DFT}}$	ACC (%)
ResNet-18	40	64	160	72.39	40	64	160	63.00	5-layer-init	40	64	160	67.51	40	64	160	54.44
			256	72.48			256	62.54				256	68.57			256	54.48
			512	73.31			512	64.85				512	69.37			512	55.95
			1024	74.88			1024	64.69				1024	69.68			1024	57.78
			1600	74.08			1600	65.25				1600	68.32			1600	56.98
		128	160	78.66		128	160	71.31			128	160	73.13		128	160	62.24
			256	78.97			256	70.66				256	72.39			256	62.53
			512	79.38			512	70.96				512	73.19			512	63.13
			1024	79.29			1024	71.67				1024	73.34			1024	62.79
			1600	78.09			1600	69.57				1600	71.96			1600	61.73
		256	160	81.25		256	160	75.69			256	160	74.97		256	160	65.09
			256	80.74			256	75.64				256	75.12			256	66.04
			512	81.94			512	75.45				512	75.15			512	66.19
			1024	81.57			1024	75.29				1024	75.25			1024	64.95
			1600	80.04			1600	74.29				1600	72.94			1600	64.17
	128	64	160	73.79	128	64	160	63.53		128	64	160	69.52	128	64	160	55.96
			256	73.47			256	63.44				256	68.50			256	56.43
			512	74.41			512	64.67				512	69.37			512	56.10
			1024	75.21			1024	65.52				1024	69.93			1024	57.46
			1600	76.44			1600	66.32				1600	69.76			1600	57.68
		128	160	78.30		128	160	71.15			128	160	73.85		128	160	62.13
			256	79.34			256	71.05				256	74.20			256	62.50
			512	79.14			512	72.00				512	73.32			512	61.79
			1024	78.42			1024	72.03				1024	72.39			1024	61.51
			1600	77.79			1600	71.09				1600	72.28			1600	60.38
		256	160	81.50		256	160	75.39			256	160	76.05		256	160	67.61
			256	80.48			256	75.24				256	76.61			256	65.83
			512	80.99			512	75.01				512	75.85			512	65.27
			1024	79.53			1024	73.96				1024	75.51			1024	64.59
			1600	80.67			1600	73.74				1600	75.40			1600	64.01

**Table 3 sensors-21-07728-t003:** Phoneme Error Rates (PER) calculated for different tested architectures and all combinations of speech preprocessing parameters on the TIMIT core test set.

Network Architecture	Central Window Method				Sliding Window Method				Network Architecture	Central Window Method				Sliding Window Method			
	$N_{\mathrm{FB}}$	WL	$n_{\mathrm{DFT}}$	ACC (%)	$N_{\mathrm{FB}}$	WL	$n_{\mathrm{DFT}}$	ACC (%)		$N_{\mathrm{FB}}$	WL	$n_{\mathrm{DFT}}$	ACC (%)	$N_{\mathrm{FB}}$	WL	$n_{\mathrm{DFT}}$	ACC (%)
ResNet-18			160	28.64			160	27.01	5-layer-init			160	33.40			160	36.94
			256	28.88			256	28.88				256	32.74			256	37.14
		64	512	27.78		64	512	27.72			64	512	31.28		64	512	37.01
			1024	26.01			1024	27.41				1024	30.87			1024	36.04
			1600	27.46			1600	27.75				1600	33.74			1600	36.17
			160	23.02			160	23.28				160	28.12			160	31.71
			256	22.84			256	23.74				256	28.57			256	31.95
	40	128	512	21.96	40	128	512	24.12		40	128	512	28.27	40	128	512	31.84
			1024	22.61			1024	24.14				1024	27.93			1024	32.83
			1600	24.59			1600	24.53				1600	29.36			1600	34.32
			160	19.73			160	21.11				160	25.84			160	30.71
			256	20.07			256	20.54				256	26.41			256	29.72
		256	512	19.33		256	512	21.76			256	512	26.23		256	512	30.71
			1024	19.75			1024	21.99				1024	26.46			1024	31.13
			1600	21.61			1600	23.46				1600	28.22			1600	32.00
			160	27.93			160	28.64				160	31.67			160	35.17
			256	27.34			256	27.17				256	32.45			256	36.57
		64	512	27.30		64	512	27.83			64	512	32.19		64	512	36.61
			1024	25.18			1024	25.26				1024	31.28			1024	35.90
			1600	25.14			1600	25.48				1600	31.86			1600	34.65
			160	22.31			160	23.09				160	27.78			160	31.86
			256	22.51			256	22.70				256	27.30			256	30.79
	128	128	512	21.77	128	128	512	22.37		128	128	512	28.33	128	128	512	32.19
			1024	24.34			1024	23.30				1024	29.26			1024	32.82
			1600	24.99			1600	24.53				1600	29.66			1600	35.10
			160	20.55			160	21.87				160	24.84			160	28.37
			256	20.55			256	21.91				256	24.40			256	31.12
		256	512	21.21		256	512	19.96			256	512	24.58		256	512	30.96
			1024	18.67			1024	23.05				1024	25.35			1024	31.30
			1600	20.05			1600	22.60				1600	26.71			1600	32.08

**Table 4 sensors-21-07728-t004:** Accuracy of the optimized 5-layer network in spectrogram-window classification (ACC) as well as in phoneme recognition (PER).

Network Architecture	Central Window Method					Sliding Window Method				
	$N_{\mathrm{FB}}$	WL	$n_{\mathrm{DFT}}$	ACC (%)	PER (%)	$N_{\mathrm{FB}}$	WL	$n_{\mathrm{DFT}}$	ACC (%)	PER (%)
5-layer-opt	40	256	160	80.41	23.68	40	256	160	72.23	29.22
			256	80.13	23.87			256	72.64	27.83
			512	79.61	24.46			512	72.70	27.31
			1024	79.93	24.37			1024	70.69	29.22
			1600	78.81	25.78			1600	70.63	29.33
	128	256	160	81.25	23.58	128	256	160	72.86	26.80
			256	80.91	23.42			256	72.56	27.60
			512	81.24	22.91			512	72.70	26.51
			1024	80.78	23.01			1024	72.28	27.36
			1600	80.19	23.69			1600	71.96	26.37

**Table 5 sensors-21-07728-t005:** Performance of classifier ensembles (ResNet-18 and the simple unoptimized CNN).

Network Architecture	Central Window Method			Sliding Window Method			Network Architecture	Central Window Method			Sliding Window Method		
	$N_{\mathrm{FB}}$	WL	PER (%)	$N_{\mathrm{FB}}$	WL	PER (%)		$N_{\mathrm{FB}}$	WL	PER (%)	$N_{\mathrm{FB}}$	WL	PER (%)
ResNet-18	40	64	24.73	40	64	25.36	5-layer-init	40	64	29.70	40	64	34.61
		128	20.04		128	21.23			128	25.84		128	29.56
		256	17.32		256	18.60			256	23.92		256	27.45
	128	64	24.33	128	64	23.57		128	64	29.98	128	64	34.35
		128	20.15		128	20.18			128	25.37		128	29.80
		256	18.06		256	18.68			256	22.12		256	28.15

**Table 6 sensors-21-07728-t006:** Required challenge length (*n*—in phonemes) and the minimum number of correct matches (*k*) for different assumed presentation-attack detection confidence levels and selected architectures.

Network Architecture	Assumed PAD Confidence		
	0.90	0.95	0.99
5-layer-init CNN	n=5,k=4	n=6,k=3	n=10,k=4
Ensemble of 5-layer-init CNNs	n=5,k=3	n=6,k=3	n=9,k=4
ResNet-18	n=5,k=3	n=6,k=3	n=9,k=4

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
