# Peer review of "Presentation Attack Detection on Limited-Resource Devices Using Deep Neural Classifiers Trained on Consistent Spectrogram Fragments"

_sensors, 2021, doi:10.3390/s21227728_

Round 1

Reviewer 1 Report

Dear authors, the article is very interesting,  the utilisation of mobiles  or embedded devices are growing in the world. So the proposed training scheme for develop reliable challenge-response based presentation-attack detection algorithms  , which utilize neural architecture is very important.The the proposed concept can become an good alternative to other existing PAD methods for voice-based user-authentication.

Reviewer 2 Report

This paper proposes a method of presentation attack detection based on simplified neural network architecture and consistent spectrogram fragments. The experimental results show the correctness and effectiveness of the proposed approach. This manuscript and it can be revised in the following aspects.

  1. The title of the article is too long to be a suitable one.
  2. In the experimental section, in Tab. 3 and 5, the proposed method cannot outperform other methods in all situations. How to explain and improve the proposed model? The comparison with the state-of-the-art methods on limited-resource devices are expected. It is necessary to supplement the computation complexity, execution time and memory requirements of the proposed method.

Reviewer 3 Report

  • It is unclear from the manuscript what the authors have done in this work. The title starts with “Boosting performance ~”, but no information about the boosting has been found from the manuscript. The proposed method section describes mainly about three data processing phases.
  • The manuscript is difficult to follow. Particularly, the authors tend to have long sentences which hinder readership. Moreover, the manuscript contains Grammar errors here and there over the entire manuscript.
  • Based on these observations above, I would like to say that the current manuscript is not ready to be considered for a journal publication.

Round 2

Reviewer 3 Report

  • Thanks for the authors’ efforts. The revised manuscript seems better than the initial version in terms of presentation. However, I am still thinking that the manuscript needs another round of major revision to be considered for a journal publication in Sensors. Please find my comments below:
  • It is unclear to me what are the authors’ contributions in the work. What I have understood from the Introduction section is that a central-window scheme is proposed to replace the conventional sliding-window scheme which is utilized in generating spectrograms. With the central-window scheme, the authors claimed that training a CNN can be lightened. If my understanding has no flaws, the current manuscript is not able to show the authors’ main idea.
  • Continued to the above, the proposed method section (I guess Section 3) is insufficient. Except for Section 3.2, Section 3 describes the conventional procedure of Mel-spectrogram coefficients (including Fig. 1). The title of Section 3.2 is misleading also.
  • The title of the manuscript is also overclaimed.
